# The Effect of Boreal Summer Intraseasonal Oscillation on Evaporation Duct and Electromagnetic Propagation over the South China Sea

**Wentao Jia [1,2], Weimin Zhang [2,\*], Jiahua Zhu [2] and Jilin Sun [1,\*]**

[1]  College of Oceanic and Atmospheric Sciences, Ocean University of China, Qingdao 266100, China; nudt_jwt@163.com

[2]  College of Meteorology and Oceanography, National University of Defense Technology, Changsha 410073, China; zhujiahua1019@hotmail.com

\*  Correspondence: weiminzhang@nudt.edu.cn (W.Z.); rainbetimes@163.com (J.S.); Tel.: +86-0731-87021601 (W.Z.); +86-0532-66782556 (J.S.)

**Abstract:** Intraseasonal oscillation of the evaporation duct, lasting 30–60 days, has been identified over the South China Sea (SCS) summer monsoon region based on multiple reanalyses and observational data. The boreal summer intraseasonal oscillation (BSISO) causes anomalies at the air–sea boundary and thus plays a dominant role in modulating the variation of the evaporation duct. The height and strength of the duct enhance/suppress during the negative/positive phase of the BSISO over the SCS. This results from the fact that active BSISO convection reduces solar radiation reaching the sea surface by increasing cumulus cloud cover, whereupon precipitation and water vapor transported by the enhanced southwest jet increase humidity over the air–sea boundary. Reduced air–sea temperatures and humidity differences lead to a weaker evaporation duct. Usually, the temporal evolution of the evaporation duct lags 2–4 days behind the BSISO, with the center of evaporation duct anomalies farther south than the BSISO. Simulated electromagnetic fields substantively influence the condition of the evaporation duct, with obvious over-the-horizon and radar blind spot effects in the typical negative phase of the BSISO, which is very different from standard atmospheric conditions.

**Keywords:** boreal summer intraseasonal oscillation; evaporation duct; P-J model; electromagnetic wave

## 1. Introduction

The atmospheric duct is a type of atmospheric junction formed in the troposphere, with a significant impact on the propagation of electromagnetic waves. According to the physical mechanism and height of the duct, it can be divided into three parts: surface duct, elevated duct, and evaporation duct [1]. The evaporation duct is a surface duct caused by evaporation from the sea surface, and is mainly formed by sharp decreases in humidity at the air–sea boundary. Generally, the height of the evaporation duct is less than 40 m [2]. It is the kind of atmospheric duct with the highest probability of occurrence and the greatest impact on electronic equipment in the ocean. Consequently, it has wide significance, particularly for the military, and much research has sought to understand its fundamental principles. The evaporation duct is usually characterized by significant diurnal and seasonal variations: generally, for the same area of sea, the height of the evaporation duct is higher during the day than at night, and higher in summer than in winter.

The variation in the evaporation duct is complex, since it is very sensitive to hydrometeorological factors at the air–sea boundary, including wind speed, sea surface temperature (SST), and humidity [2–7]. Usually, it is difficult to obtain the necessary observational data along a vertical gradient above the sea surface, particularly in the open sea. Therefore, evaporation duct models based on Monin–Obukhov

similarity theory and statistical rules are widely used, such as the P-J model [3,4], MGB model [5], Babin model [6], and others. In recent years, with the development of computer technology, optimized models based on machine learning have been put forward [7,8]. These enable the acquisition of vertical refractivity profiles close to the surface of the ocean, which are necessary to predict radar and communications performance by microwave models, such as the Electromagnetic Parabolic Equation Routine.

According to statistical analyses, the probability of an evaporation duct in the South China Sea (SCS) is close to 100%, and this phenomenon exhibits significant seasonal differences [9]. The boreal summer intraseasonal oscillation (BSISO) is one of the dominant components of tropical atmospheric low-frequency oscillations and is mostly active as a northward propagating feature in the Asian summer monsoon region [10–15]. The highest low-frequency variability in BSISO activity occurs in the tropical Western North Pacific (WNP), including the SCS. During the summer, when the 30–60-day BSISO propagates northeastward across the SCS, intraseasonal variation of meteorological factors are observed along the propagation pathway from the equator to the Northwest Pacific, including outgoing long wave radiation (OLR), sea surface temperature (SST), evaporation, rainfall, and wind [10–20]. Previous studies have revealed that sea surface evaporation correlates closely with Madden-Julian oscillation (MJO) and BSISO, usually lagging behind enhanced convection by 1–2 weeks in the equatorial Indian and western Pacific oceans, leading us to believe that BSISO activities influence the evaporation duct in this way [17–20]. In this study, we aimed to reveal the effects of BSISO on the evaporation duct over the SCS. We also present general features of electromagnetic propagation in the duct environment.

## 2. Data and Methods

### 2.1. Data

Daily mean 10-m U and V wind, humidity, temperature, and SST data at 0.125 degrees were obtained from ERA-Interim datasets [21], and were used to calculate the evaporation duct according to the P-J model. Daily mean outgoing longwave radiation (OLR) data at 1 degree were obtained from the National Oceanic and Atmospheric Administration (NOAA) datasets [22]. The third version of global sea surface flux products developed by the Objectively Analyzed air-sea Heat Fluxes (OAFlux) project at the Woods Hole Oceanographic Institution (WHOI) was also used in this study [23]. The products of OAFlux are the sum of surface meteorological variables, including shortwave (SWF) and longwave (LWF) radiative fluxes, ocean evaporation, heat fluxes, and other factors on daily and 1 degree scales. A time period of 1996–2016 was applied for all the datasets used. Additionally, we used precipitation based on Tropical Rainfall Measurement Mission (TRMM) [24] and real-time observation data (including evaporation duct height and other meteorological factors) obtained from the Chinese Navy.

### 2.2. BSISO Analysis

A number of studies have demonstrated that the BSISO over the SCS has two significant periods of intraseasonal oscillation, one of approximately 10–25 days and another of around 30–60 days. To extract BSISO signals, a bandpass filter was applied to daily anomalies (removing climatology from the daily mean) of variables, including OLR, wind, and precipitation [11,19]. We chose the Lanczos bandpass filter to process 21 years of OLR data by virtue of its advantages for computing long timescales [25,26]. Figure 1a–c shows 30–60-day bandpass-filtered OLR anomalies based on empirical orthogonal function (EOF) analysis, representing the location of deep convection in BSISO phases. These three EOF modes explain approximately two-thirds of the total variance (EOF1, 30.3%; EOF2, 18.8%; EOF3, 9.9%). This shows that BSISO-related convection originates from the equatorial western Pacific, strengthens at around 15° N, and dissipates over subtropical mainland China. Cross-correlation between the first and second principal components (PC1 and PC2, i.e., the solid red line in Figure 1d) is minimized

(maximized) when PC2 lags (leads) PC1 by approximately 10 days. Principal components PC2 and PC3 (the solid blue line in Figure 1d) suggest that the lifecycle of the BSISO is around 40–60 days.

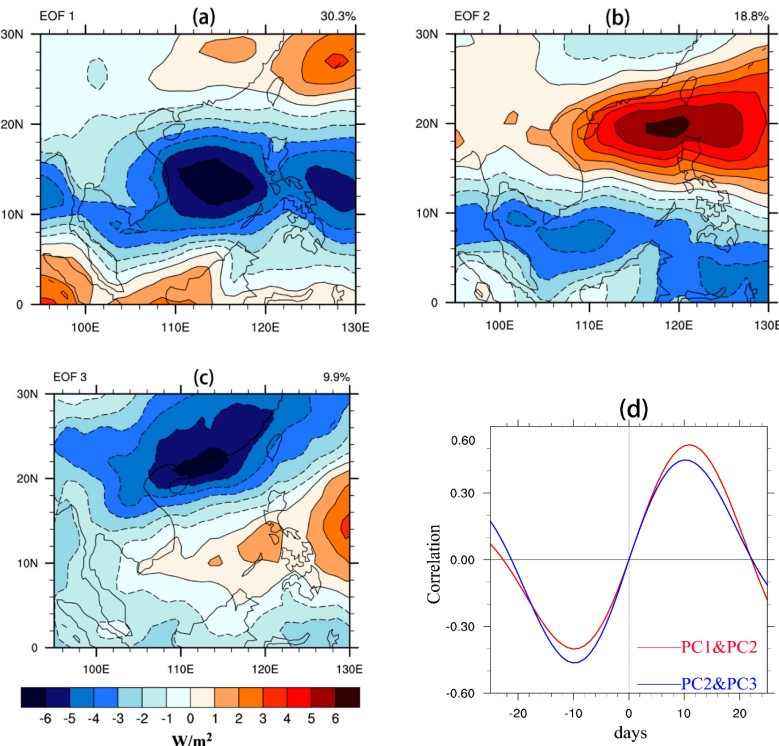

**Figure 1.** Three leading empirical orthogonal function (EOF) modes (**a**) EOF1, (**b**) EOF2, and (**c**) EOF3 of 30~60-day bandpass filtered daily OLR anomalies over the SCS for 1996~2016; (**d**) is the cross-correlation of PC1-PC2 (red solid line) and PC2-PC3 (blue solid line).

## 2.3. Paulus-Jeske (P-J) Model

Generally, evaporation duct models aim to determine the expression of the vertical refractivity gradient in terms of atmospheric variables. Since the evaporation duct is limited to the surface layer, Monin–Obukhov similarity theory (M-O theory) is useful for deriving this expression [27]. The P-J model is likely the most widely used and successful evaporation duct prediction model, and has been used, for example, in U.S. Navy electromagnetic propagation business software systems (IREPS and AREPS) since 1978. The P-J model uses a pressure of 1000 hPa, potential temperature instead of absolute temperature, and potential water vapor pressure instead of absolute water vapor pressure to calculate potential refractivity [3,4]. It also uses the bulk Richardson number to categorize atmospheric stability and estimate the Obukhov length. Paulus (1989) [28] derived the critical potential refractivity gradient for ducting as Equation (1):

$$\frac{\partial N_p}{\partial z} = \frac{\partial N}{\partial z} - \frac{\partial N}{\partial p}\frac{\partial p}{\partial z},$$ (1)

where $N_p$ is potential refractivity and $N$ is nonpotential refractivity. The term $\partial N/\partial z$ is set at the critical gradient of −0.157 for ducting. Using the hydrostatic relationship and the ideal gas law, Paulus obtained $\partial p/\partial z = -0.12$. Based on these values, the critical potential refractivity gradient for ducting is −0.125.

The derivation process of duct height is beyond the scope of this study, so we give the expression directly as follows:

$$Z_{DH} = \begin{cases} \dfrac{\Delta N_p}{b_c B_1 - \Delta N_p(\alpha_1/L)} \\[2em] \left[ \left(\dfrac{b_c B_2}{\Delta N_p}\right)^4 - \dfrac{4\alpha_2}{L}\left(\dfrac{b_c B_2}{\Delta N_p}\right)^3 \right]^{-\frac{1}{4}} \end{cases} , \tag{2}$$

where $Z_{DH}$ is the height of the evaporation duct, $\Delta N_p$ is the difference in potential refractivity between the measured altitude and sea level, and $b_c$ is the critical potential refractivity gradient. Under a stable atmosphere, $B_1 = \ln[(z/z0) + z(\alpha 1/L)]$, aerodynamic roughness $z_0 = 1.5 \times 10^{-4}$ m, constant $\alpha_1 = 5.2$, and $L$ is the Obukhov length. Under unstable conditions, $B_2 = \ln[(z/z0) - \varphi]$, and $\varphi$ is calculated empirically.

Herein, we test whether the P-J model is applicable for the prediction of evaporation ducts over the SCS in summer. We used real-time evaporation duct data from the Chinese Navy for summer 2016, and the voyage route shown as Figure 2a. During this voyage, we acquired more than 1000 samples over the SCS, 200 of which were under different stable conditions after the removal of errors and single-station duplicate data. A comparison between our results shows that both under stable (Figure 2b) and unstable (Figure 2c) atmospheric conditions, the heights of the evaporation duct predicted by the P-J model are usually lower than observations, especially for unstable conditions. The results of the test data indicate that the P-J model is reliable, and that its root of mean square error (RMSE) is less than 20%. In conclusion, the error range of the P-J model satisfies the needs of analyzing low-frequency changing trends in the evaporation duct over the SCS.

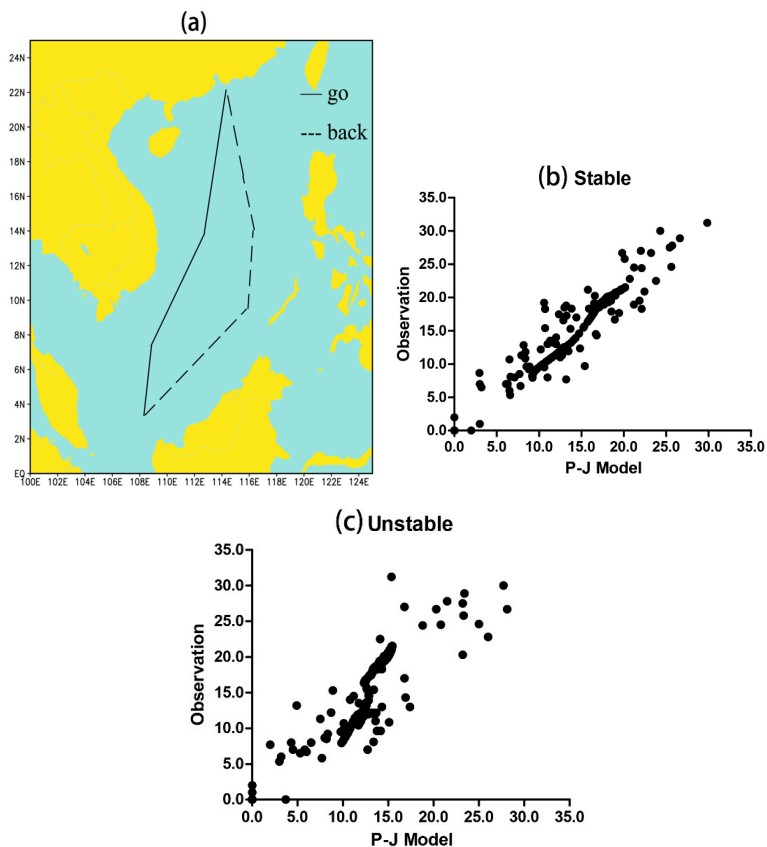

**Figure 2.** Accuracy test for the 2.3. Paulus-Jeske (P-J) model over the South China Sea (SCS) region during summer: (**a**) the voyage route of the Chinese Navy ships in summer 2016; (**b**,**c**) comparison of P-J models and observed duct height in (**b**) stable and (**c**) unstable atmospheric conditions.

### 3. ISO of Evaporation Duct

As mentioned above, evaporation ducts result from a sharp decrease of humidity at the air–sea boundary, mostly due to sea surface evaporation. By using 21 years (1996–2016) of daily anomaly data to compute the ratio of bandpass variance to unfiltered variance, we obtained the variance of BSISO-related sea surface evaporation (Figure 3a,b). In summer, lower tropospheric evaporation in the northern hemisphere was higher than that in the southern hemisphere, whereas the opposite was true in winter, which is similar to seasonal features of MJO. There were two most significant areas in summer: the first was the western North Pacific Ocean, especially in the southern SCS area, and the other the northern Indian Ocean. These areas correspond to the active regions of BSISO during summer.

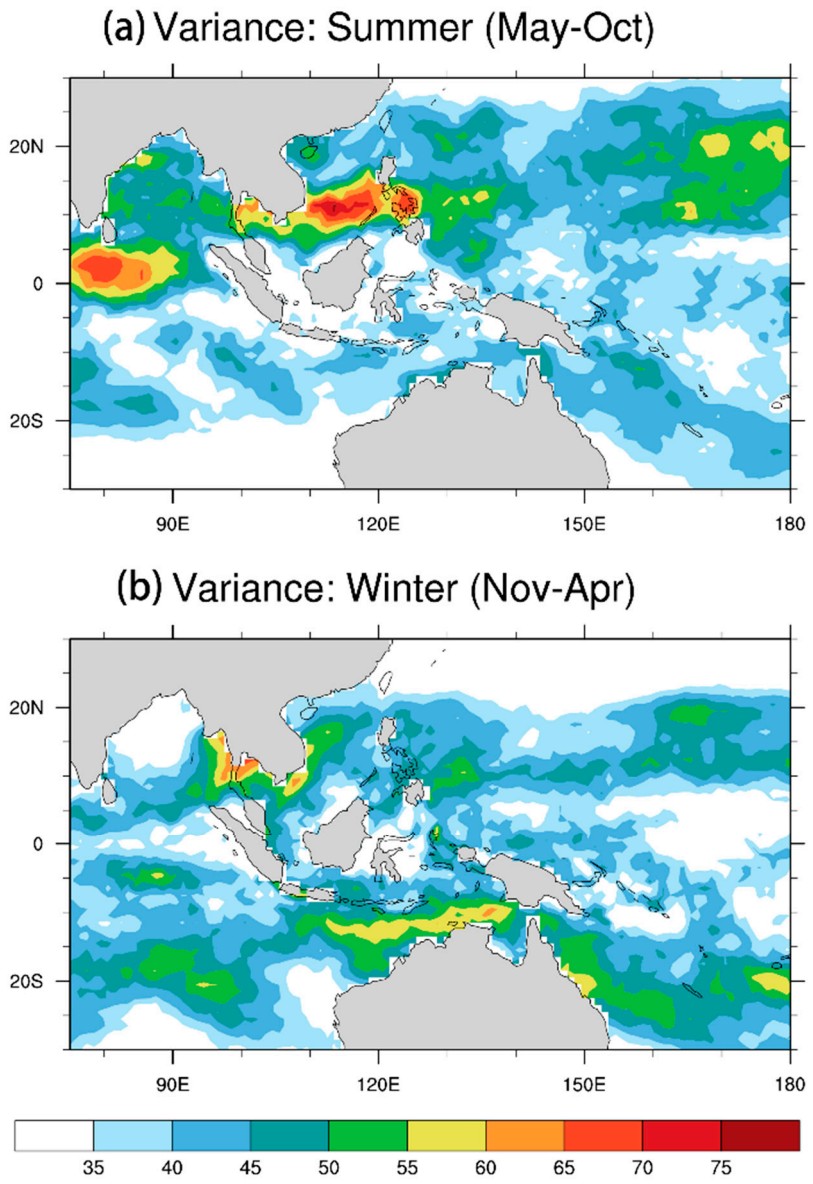

**Figure 3.** Variance of the boreal summer intraseasonal oscillation (BSISO)-related lower tropospheric evaporation in (**a**) summer (May–October) and (**b**) winter (November–April) based on OAFlux data (1996–2016).

Lower tropospheric evaporation in the SCS shows the most prominent variability on intraseasonal timescales. However, sea surface evaporation does not equate to the evaporation duct, thus we require further verification of BSISO-related characteristics. The two dominant modes of the BSISO over the

WNP with different frequency bands are those corresponding to 30–60 days and 10–25 days, of which the former is more significant [12,20]. The power spectra of OLR anomalies over the SCS (Figure 4a) are concentrated in two intraseasonal periods similarly to BSISO cycle time, as are the power spectra of the evaporation duct (Figure 4b). These confirm that the evaporation duct over the SCS has an intraseasonal variation in summer with the life cycle of 30–60 days. Thus, it is possible to further prove that BSISO activities may have influence the evaporation duct in the SCS area.

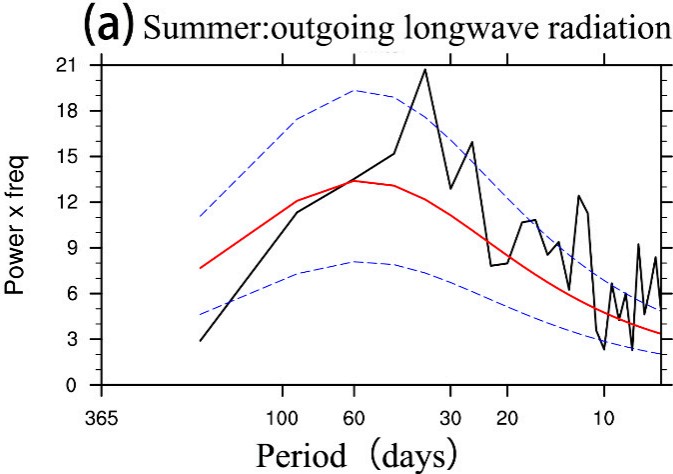

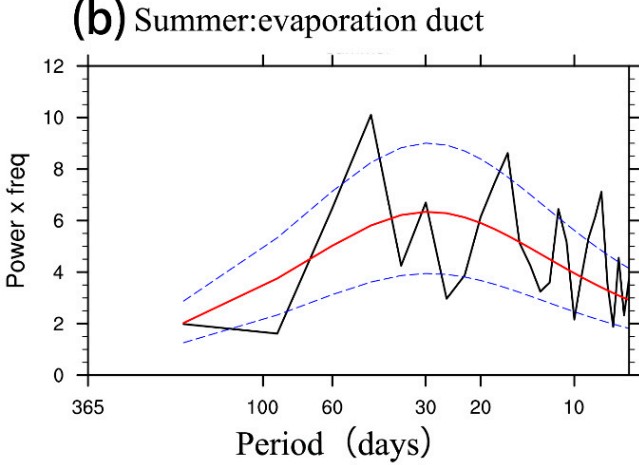

**Figure 4.** Power spectra of BSISO-related anomalies of the lower-tropospheric (**a**) OLR and (**b**) evaporation duct over the SCS in summer. The red curve represents the red-noise spectrum. The lower and upper blue dashed curves are the 5% and 95% red-noise significance levels, respectively.

## 4. ISO of Evaporation Duct

### 4.1. Correlation Analysis

We obtained the three leading modes and time coefficient of the BSISO based on the EOF analysis in Section 2. In order to elucidate the relative effect of the BSISO on evaporation ducts, we calculated the correlation between the time coefficient of the EOF modes and duct height anomalies (Figure 5). This shows that the evaporation duct is closely correlated with BSISO-related OLR anomalies, and that enhanced/suppressed convection is accompanied by negative/positive duct height anomalies.

In phases 1 and 2 of the BSISO, the maximum correlation coefficient is greater than 0.5, i.e., passing the t-test. Variations in duct intensity also have similar characteristics (figure not shown). However, slight inconsistencies remain between the locations of BSISO-related OLR and evaporation ducts, since the center of duct height anomalies is almost 5 latitude degrees southward. In other words, the weakening trend of evaporation duct lags behind BSISO-related deep convection (negative OLR anomalies) over the SCS during the summer. The specific reasons for this are explained in Section 5.

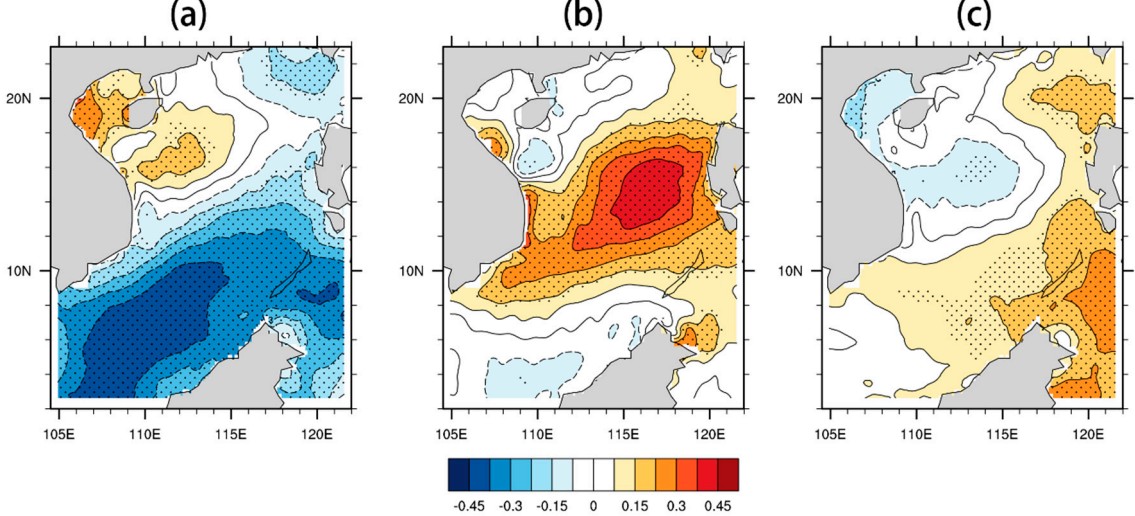

**Figure 5.** Correlation coefficients of BSISO-related anomalies of evaporation duct height over the SCS in summer. (**a**–**c**) correspond to the first/second/third EOF modes in Section 2. Dotted areas are statistically significant at the 99% confidence level based on a significant t-test.

## 4.2. Composite Analysis

Figure 6 shows the composite 30–60-day bandpass-filtered anomalies of OLR, wind, and the evaporation duct during active convection phases of the BSISO over the SCS, which are identified via the principal components of EOF analysis in Section 2. We divided the northward propagation from the equator to subtropical East Asia into five phases accompanied by obviously cyclonic low-level wind anomalies (concentrated on days −10, −5, 0, +5, and +10). The right panels of Figure 6 show the associated evaporation duct anomalies, exhibiting a significant northward pattern related to convection in each BSISO phase. As negative OLR anomalies (deep convection) propagate northward from phase 1 to 5, negative duct height anomalies also gradually propagate to the subtropical coast in the same manner. Compared to OLR anomalies, the center of the evaporation duct lies to the southeast of the well-developed convection. Consequently, negative duct height anomalies appear after enhanced convection, probably indicating that variations in evaporation duct require a response time to BSISO activities. The suppressed phase follows similar rules with opposite features to the evaporation duct. In conclusion, the enhanced/suppressed evaporation duct followed a synchronized change rule with negative/positive BSISO convection (positive/negative OLR anomalies) during summer in the SCS.

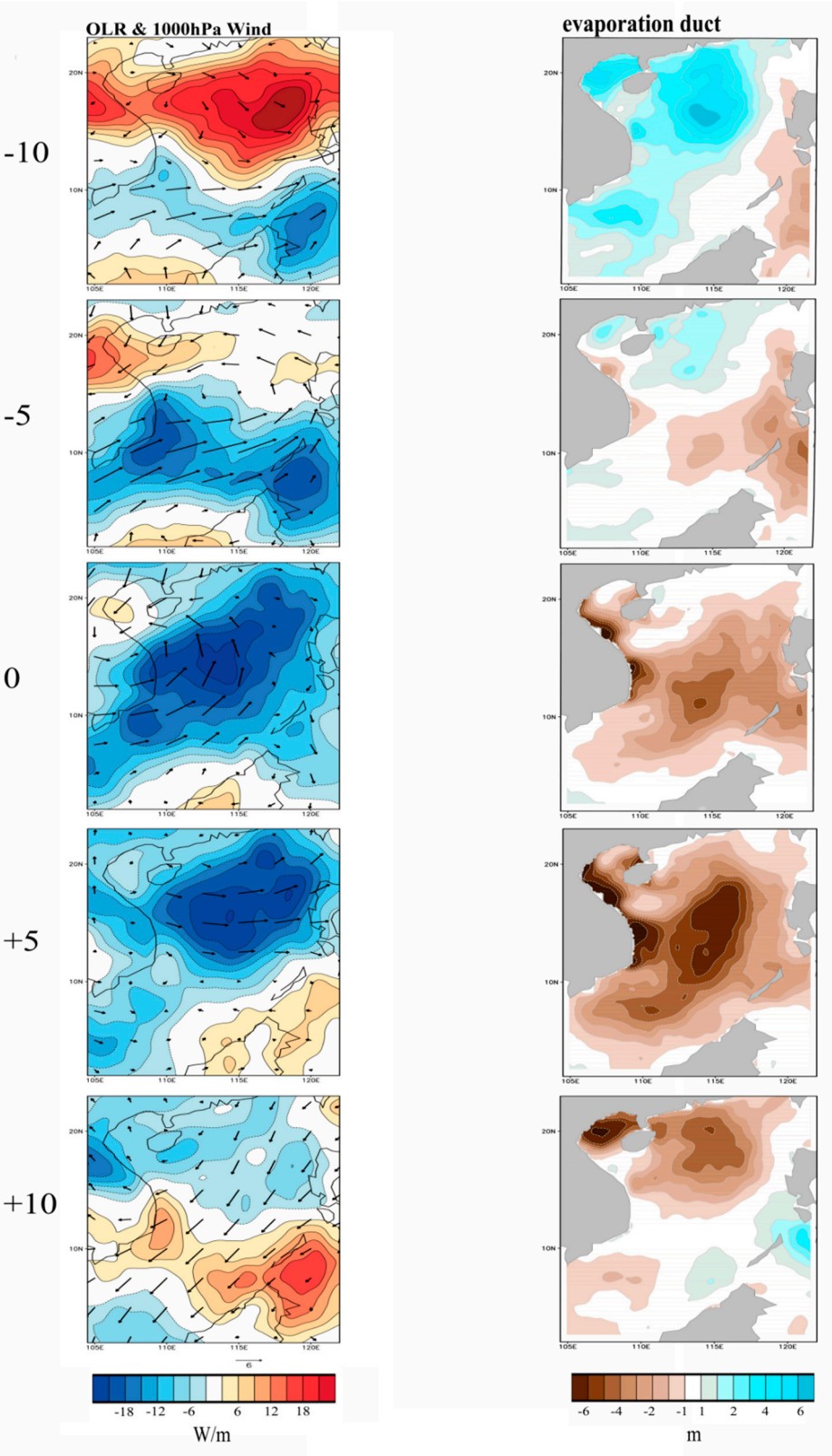

**Figure 6.** Composites of BSISO-related OLR anomalies, 1000 hPa horizontal wind vector (**left**), and evaporation duct (**right**) centered at days −10, −5, 0, +5, and +10, in terms of BSISO active events. Day 0 indicates the day at which deep convection of BSISO is maximized over the SCS, and positive (negative) days indicate the days after (before).

## 5. The Effect of BSISO on Evaporation Duct

### 5.1. Meteorological Factors Influencing the Evaporation Duct

Sea surface evaporation is predominantly influenced by wind speed, air-sea humidity, and temperature differences [29]. Evaporation processes also affect meteorological factors due to the exchange of water vapor and heat flux, which is known as wind-evaporation-SST (WES) feedback [30]. Seasonal cycles of sea surface evaporation in the SCS exhibit a "double-peak curve" as a result of the SCS monsoon onset. During winter, evaporation in the northern SCS is greater than that in the southern SCS due to strong winds associated with the winter monsoon. During the summer monsoon, evaporation in the central and southern SCS begins to exceed the northern part, with maxima at 8–15° N [31].

Due to the lack of high-density long-term observation data, satellite remote sensing data are most commonly used to estimate sea surface evaporation based on empirical equations. For example, the Tropical Ocean Global Atmosphere Coupled Ocean-Atmosphere Response Experiment (TOGA-COARE) used the COARE 3.0 equation for sea surface evaporation as follows:

$$Evp = \rho_a C_e U (q_s - q_a), \tag{3}$$

where Evp is evaporation, $\rho_a$ is air density, $C_e$ is the moisture and heat transfer coefficient, $U$ is wind speed at a reference height (10 m), $q_s$ is the saturation-specific humidity in sea surface temperature (SST), and $q_a$ is specific humidity at a reference height [32,33]. In the OAFlux dataset, the expression of the evaporation equation is analogical, based on bulk aerodynamic formulae. The expression is as follows:

$$Evp = Q_{LH} / \rho_w L_e, \tag{4}$$

where $\rho_w$ is the density of sea water, and $L_e$ is the latent heat of vaporization, expressed as $L_e = (2.501 - 0.00237 \times SST) \times 1.06$. $Q_{LH}$ is the respective latent heat flux for which the expression is $Q_{LH} = \rho L_e C_e U (q_s - q_a)$, where $\rho$ is the density of air, $L_e$ is the latent heat of evaporation, U is wind speed relative to the sea surface at a reference height, $C_e$ is the turbulent exchange coefficients for latent heat fluxes, and specific humidities are denoted by $(q_s - q_a)$ [23].

Sea surface evaporation cannot entirely reflect the strength of the evaporation duct, thus we need to further explore the specific effects of different factors on the evaporation duct. In this study, three groups of control experiments based on the P-J model were designed to verify the effects of wind speed, air-sea humidity, and temperature differences. The experimental settings are shown in Table 1.

**Table 1.** Control experiment settings.

|       | Wind      | Humidity  | Air Temperature | Sea Surface Temperature | Pressure  |
|-------|-----------|-----------|-----------------|-------------------------|-----------|
| exp-1 | 5 m/s     | 100%~70%  | 25.5 °C         | 25 °C                   | 1015 hPa  |
| exp-2 | 5 m/s     | 80%       | 25~28 °C        | 25 °C                   | 1015 hPa  |
| exp-3 | 1~12 m/s  | 80%       | 25.5 °C         | 25 °C                   | 1015 hPa  |

Figure 7a shows that evaporation height has a significant negative correlation with atmospheric humidity. This is because the mechanism of the evaporation duct involves a sharp decrease in humidity over a small vertical spatial range at the air–sea boundary, and because lower air humidity corresponds to a higher vertical rate of decline of water vapor. Figure 7 also demonstrates that the influence of atmospheric humidity on duct height is more influential than other meteorological factors. Temperatures at the reference height changed from 25 to 28 °C, i.e., the air–sea temperature difference changes from 0 to 3 °C; the resulting variation in evaporation duct height is shown in Figure 7b. This indicates that, when the temperature difference is less than 1.6 °C, the duct is enhanced with increasing temperature difference; however, when the temperature difference is greater than

1.6 °C, the evaporation duct will continually decline with increasing temperature difference. In the last experiment, horizontal wind speed at the reference height changed from 1 to 12 m/s, and the corresponding variation in the evaporation duct is shown in Figure 7c. When the wind speed is less than 3 m/s, the results become distorted under this assumed condition. At wind speeds greater than 3 m/s, the duct height decreases with increasing horizontal wind speed, and the rate of decrease gradually levels off, even as wind speed continue to rise.

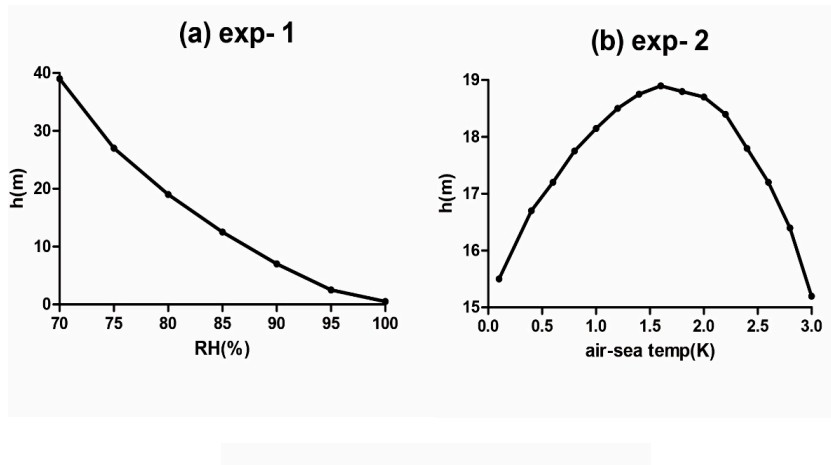

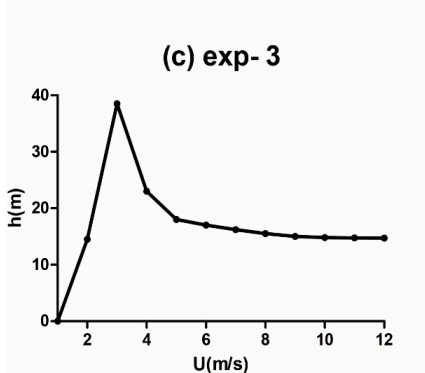

**Figure 7.** Influence of (**a**) air humidity, (**b**) air–sea temperature difference, and (**c**) horizontal wind speed on evaporation duct height based on the Paulus-Jeske (P-J) model.

Among these three factors, the greatest influence on the evaporation duct is air humidity, followed by air–sea temperature difference, then horizontal wind speed. However, Figure 7 shows that the variation in evaporation duct height is only caused by a single factor, regardless of other meteorological factors. Indeed, the evaporation duct is not influenced solely, or even mostly, by air humidity. In real air-sea conditions, the results are thus expected to be more complicated.

*5.2. Schematic Analysis*

In the previous section, we demonstrated that the principal factors affecting evaporation duct height are air–sea humidity and temperature. We now explore how the BSISO affects these factors. The latitude-based lead-/lag-time diagram of the OLR, humidity, and air-sea temperature anomalies along the section from 110° E to 130° E is shown in Figure 8. Several features are visible in the results: (1) organized intraseasonal oscillations in OLR, humidity, and air–sea temperature anomalies propagate northward over the SCS; (2) the temporal evolution of air–sea temperature and humidity both lag behind the OLR, in which enhanced (suppressed) convection leads to negative (positive) temperature and humidity differences, which can explain why the center of the evaporation duct anomalies is further south than BSISO convection (cf. Figures 5 and 6); (3) temperature is seemingly more sensitive to BSISO convection than humidity, with the temporal evolution of temperature anomalies being

approximately 4 days in advance; (4) in terms of spatial distribution, the center of humidity difference is approximately 2 latitude degrees to the north of the OLR while the center of temperature difference is about 4 latitude degrees to the south. For this reason, BSISO-related phases of air humidity and air–sea temperature are exactly in the same space-time position.

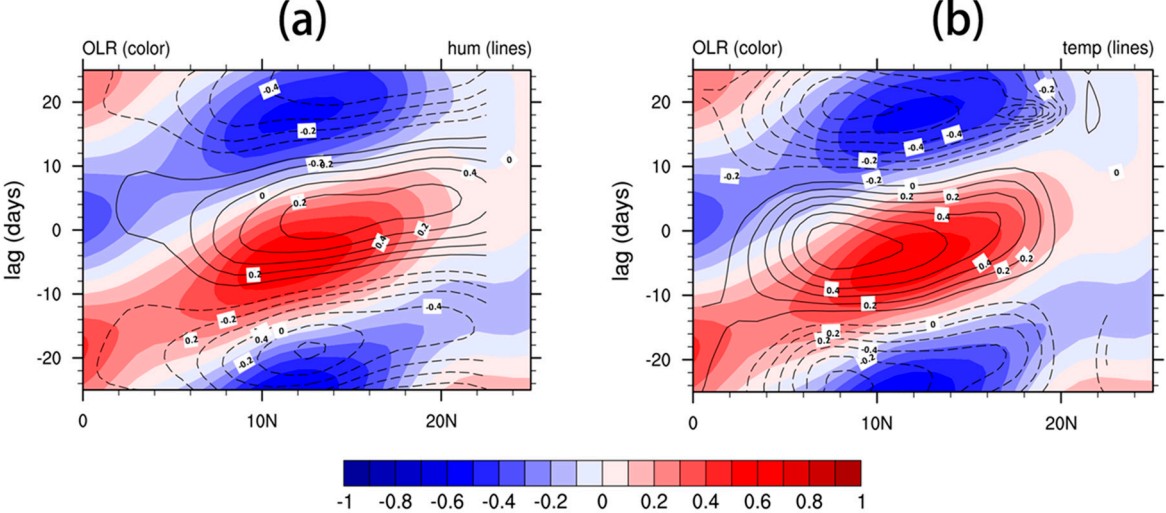

**Figure 8.** Latitude-based lead-/lag-time diagram of the composite raw pentad anomalies along the section of 110° E to 130° E for (**a**) outgoing longwave radiation (OLR, shaded) and air humidity (contour) and (**b**) OLR (shaded) and air-sea temperature (contour). The lead-/lag-time in days with positive (negative) values indicates the number of days after (before) day 0, equivalent to that shown in Figure 6.

During summer, the air temperature over the SCS is predominantly influenced by solar radiation, whereas variation in air humidity results from precipitation, water vapor transport by monsoon, evaporation, and other factors. Based on the PC1 coefficients of EOF models, BSISO-related anomalies of downward short-wave radiation flux (SWF), upward long-wave radiation flux (LWF), and SST are shown in Figure 9a. Radiation flux anomalies are highly correlated with OLR anomalies (gray shaded), indicating that the former are modulated by convection. In the case of short-wave radiation, due to increased cumulus and cloud top in the phase of enhanced convection, incident solar radiation is reflected and absorbed, and less short-wave radiation reaches the boundary layer. In contrast, the magnitude of long-wave radiation is much smaller than that of short-wave radiation. SST and OLR are almost in quadrature, such that enhanced (suppressed) convection leads the cold (warm) SST change by approximately a quarter of a cycle, reflecting the delay in temperature variation of the underlying surface. Since the specific heat capacity of seawater is very large, the range of variation in SST is much smaller than that of air temperature. During the summer, the ocean provides a "cold source", such that the air–sea temperature difference is positive when the air temperature rises in fair weather.

Figure 9b shows BSISO-related anomalies of precipitation (Pre), horizontal wind speed (U), and moisture flux divergence (MFD). As seen in Figure 9a, these three variables are also closely related to OLR anomalies. Active convection is always accompanied by low-level cyclonic circulation anomalies, and water vapor brought by enhanced southwest air currents leads to increases in air humidity. Meanwhile, increased atmospheric instability and convective precipitation also causes water vapor to mix evenly in the vertical direction, thus reducing the difference in air–sea humidity. By this physical mechanism, the BSISO affects the evaporation duct over the SCS (Figure 10).

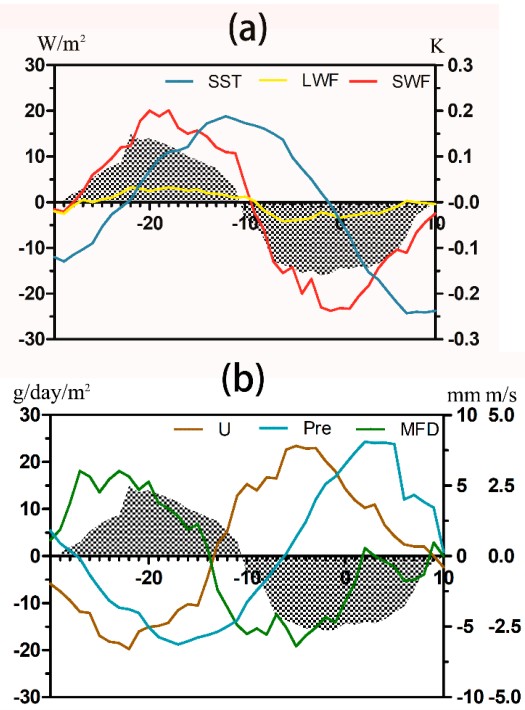

**Figure 9.** Temporal evolution of the composite original anomalies (based on the PC1 coefficient of EOF models) in the SCS region, for (**a**) downward short-wave radiation flux (SWF, red line), upward long-wave radiation flux (LWF, yellow line) and SST (blue line); (**b**) precipitation (Pre, cyan line), horizontal wind speed (U, brown line), and moisture flux divergence (MFD, green line). Gray shaded areas indicate the corresponding composite OLR anomalies, and the x-coordinate indicated the number of lead/lag days.

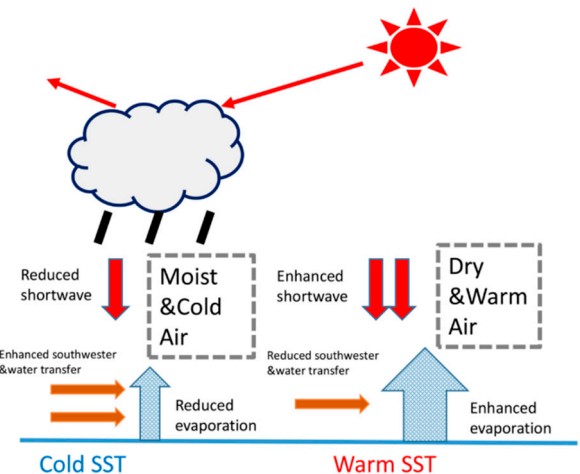

**Figure 10.** Schematic diagram showing the effect of BSISO on the evaporation duct over the SCS. Active convection induces decreased air–sea humidity and temperature differences, which leads to a weakened evaporation duct. For negative convection, the opposite is true.

## 6. The Effect of Evaporation Duct on Electromagnetic Propagation

### 6.1. Parabolic Equation

The evaporation duct has a significant effect on the propagation of electromagnetic waves. Electromagnetic waves are a kind of energy, and all bodies above absolute zero will give off radiation in

the form of electromagnetic waves. All these electromagnetic waves are governed by four mathematical equations established by James Clerk Maxwell. In this chapter, we will use observational data to simulate electromagnetic propagation under the SCS evaporation duct. The parabolic equation (PE) model is derived from Maxwell's equations and provides an approximate transmission model of electromagnetic energy along the axis. It calculates not only wave propagation beyond the visual range but also has good stability and accuracy for numerical solutions. More importantly, the PE model allows the influence of atmospheric refraction to be easily considered, such that it is widely used to determine the condition of the atmospheric duct [8,34,35]. Split-step Fourier transform (SSFT) is usually used to study the PE, the solution of which is shown in Equation (5),where u is the magnetic field, $F$ is the Fourier transform, $k$ is the wave number, $M$ is the modified refractivity, $p = k_0 \sin\theta$, and $\theta$ is the angle [36]. Thereafter, the distribution of the magnetic field along the x axis can be calculated based on the initial field, and the path loss of propagation can be obtained:

$$\mathrm{u}(x_0 + \Delta x, z) = e^{[-ik_0 \Delta x M(x,z)10^{-6}]} F^{-1}\left\{ e^{\frac{i\Delta x}{2k_0} p^2} F[u(x_0, z)] \right\}. \tag{5}$$

*6.2. Electromagnetic Simulation*

The actual observation data of evaporation ducts are relatively scarce, since they require much manpower and material resources. Data measured during cruises by scientific ships are often intermittent, and observation sites continually changing, thus such data cannot meet the requirements of this study. China has set up a meteorological gradient tower to detect evaporation duct by sensors at different heights at a weather station (9°37′ N, 112°58′ E) over the SCS area. This allows continuous and fixed-point observation data of evaporation duct to be obtained.

The BSISO is closely related to the strength of the SCS monsoon; usually, the BSISO is more visible in active monsoon years. The year 2018 was a relatively strong year in terms of the recent SCS monsoon, thus we selected hydrometeorological data from June to September 2018 for analysis. Data acquisition was conducted once per minute, but strong turbulent motion near the sea surface causes significant data fluctuation. Therefore, when cases with significant error are removed and data are averaged for each hour, better constraints of daily variations in the evaporation duct are obtained. Figure 11a shows that the evaporation duct exhibits periodic variation characteristics, consistent with the intraseasonal oscillation of the BSISO. During the negative phase of the BSISO, the averaged height of the evaporation duct is higher and exhibits significant changes, whereas in the positive phase, the height is relatively low and the variation trend more stable. We chose data typical of the BSISO positive (5–10 June) and negative (5–10 July) phases over the SCS in order to calculate modified refractivity profiles (Figure 11b,c). Figure 11b shows that the height of the evaporation duct is 6.77 m and its strength is 8.22 M, while the values shown in Figure 11c are 31.01 m and 43.56 M, respectively.

With the refractivity profiles in Figure 11, we can use the PE model to simulate the electromagnetic propagation in the condition of the evaporation duct. Besides this, the radar system parameters are also needed to input into the PE model, so as to obtain the radar detection performance predicted by the model. The radar signal parameters are shown in Table 2.

**Table 2.** Radar signal parameters.

| Frequency | Antenna | Vertical Beam Width | Elevation | Aerial Gain |
|---|---|---|---|---|
| 6 GHz | gauss | 3° | 0° | 29 dB |
| **Polarization Mode** | **Antenna Height** | **Receiver Sensitivity** | **Peak Power** | **RCS** |
| HH | 20 m | −110 dBm | 300 kW | 100 m$^2$ |

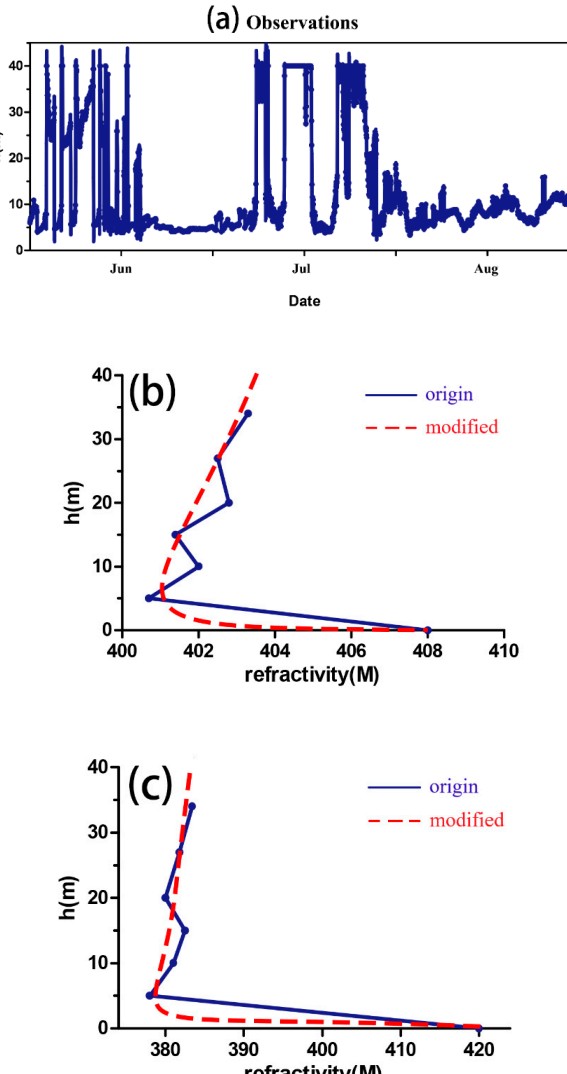

**Figure 11.** Observation data of the evaporation duct from June to September 2018 in the SCS (9°37′ N, 112°58′ E): (**a**) variation of evaporation duct height over a full BSISO cycle in the summer of 2018; (**b**,**c**) calculated refractivity profiles based on observations of the typical BSISO positive (5–10 June) and negative (5–10 July) phases during 2018.

Figure 12a,b shows the loss of electromagnetic propagation under evaporation duct conditions, where smaller values of loss denote stronger field strengths. For ease of display, we used a plane instead of Earth curves, such that the path of electromagnetic waves is bent upward. It can be clearly seen from the figure that, in the case of strong evaporation ducts, the energy of the electromagnetic wave can travel further; this is usually referred as "over-the-horizon propagation (OTHP)". Furthermore, the energy near the surface is stronger as a result of high ground or sea surface reflectivity. Although there is also a weak evaporation duct in the active convection of the BSISO, the antenna height (20 m) is greater than the duct height (6.77 m), such that the electromagnetic field does not change significantly. A stronger evaporation duct extends the range of radar detection but also forms a blind area on top of the duct (Figure 12c,d). A trapping layer can be formed from sea level to the top of the evaporation duct, causing the original electromagnetic blind area to disappear; however, the duct can change the trajectory of electromagnetic radiation, resulting in the development of a blind area (90 to 110 km) at the maximum calculated height (500 m), which is far from the starting position of the blind zone under standard atmospheric conditions.

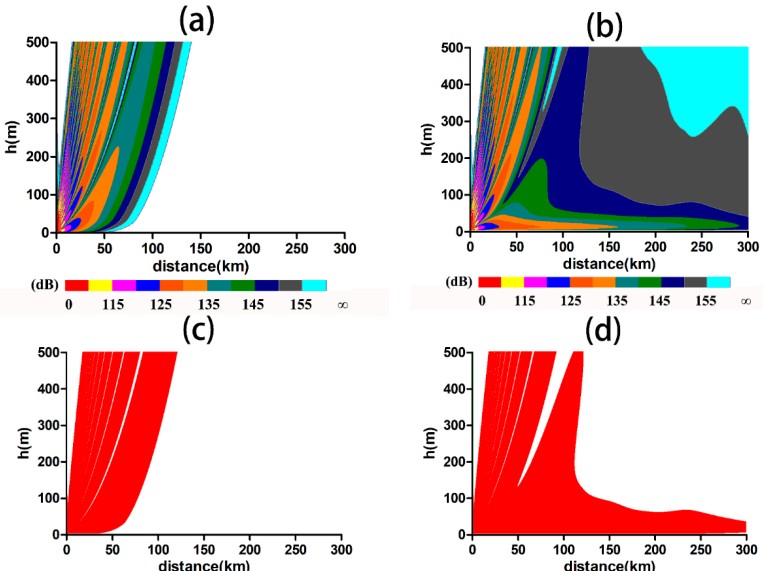

**Figure 12.** Typical traits of electromagnetic fields based on the PE model at different BSISO phases: left column corresponds to an active BSISO event, and the right corresponds to negative. (**a**,**b**) loss of electromagnetic propagation under evaporation duct conditions; (**c**,**d**) blind zone in the electromagnetic field.

## 7. Conclusions

Based on the reanalysis of data in addition to new observational data, we used the P-J model to calculate the height and strength of the evaporation duct over the SCS in summer, and analyzed the fundamental processes and mechanisms by which the BSISO influences the evaporation duct. In addition, we used PE to simulate electromagnetic propagation under evaporation duct conditions. The main conclusions of this study may be summarized as follows:

1.  The P-J model is suitable for characterizing the evaporation duct over the SCS area in summer. As a result of the BSISO, the evaporation duct exhibits an intraseasonal oscillation of 30–60 days, and shows strong correlation between certain time-space distribution features. The height and strength of the evaporation duct is enhanced/suppressed in negative/positive phases of the BSISO, leading to the development of a negative/positive center in evaporation duct anomalies to the south of the active/inactive BSISO convection. These two features do not exactly co-occur, and the evolution of evaporation duct lags behind the BSISO-related convection by about 2–4 days.
2.  Changes in the difference of temperature and humidity at the air–sea layer caused by BSISO-related convection are the dominant factors influencing the evaporation duct. Clouds, precipitation, and enhanced southwest airflow can reduce differences in air–sea temperature and humidity throughout the area of active convection, thus weakening the strength and height of the evaporation duct. Conversely, the duct will be significantly enhanced on sunny days.
3.  Based on observational data from a meteorological station in the SCS, we calculated the evaporation duct and modified refractivity profiles in typical negative and positive phases of the BSISO. The propagation of electromagnetic waves during conditions of different evaporation ducts was also simulated by the PE model. During the negative phase of BSISO convection, a strong evaporation duct causes significant over-the-horizon propagation and the development of blind areas, causing the electromagnetic fields to differ obviously from standard atmospheric conditions.

Some of the questions addressed in this study require further consideration. We used the P-J model, a mature empirical model, to calculate characteristics of the evaporation duct. Although suitable for the SCS area in summer, there are still some deviations inherent to the P-J model. In recent years, a number of new models based on machine learning have been developed to provide more precise methods and calculating procedures. Furthermore, since the BSISO reflects a low-frequency oscillation

of the atmosphere, its influence on the evaporation duct is based on long-term statistics. Although it has a certain indicative significance, the physical mechanism of the evaporation duct is very complex and influenced by many factors. As a result, the role of the BSISO may be less significant than modeled, especially when influenced by stronger weather systems in summer, such as Equatorial Easterly Wave, typhoon, etc. Besides the evaporation duct, there is also a high probability of both surface and elevated ducts in the SCS area. The influence of the BSISO on these phenomena requires further study.

**Author Contributions:** This study was conceived and designed by W.J., W.Z. and J.S. The data were analyzed by W.J. and J.Z., and the paper was written by W.J. All authors contributed to the revision of the manuscript. All authors have read and agreed to the published version of the manuscript.

**Funding:** This research was funded by the National Natural Science Foundation of China, Grant Number [41830964].

**Acknowledgments:** We are grateful to Jincai Li and Xiaojun Liu for providing us with the P-J model and PE programs. We thank the Chinese Navy for the observation data of the evaporation duct over the SCS area. We would like to thank the reviewers for their suggestions and comments.

**Conflicts of Interest:** The authors declare no conflict of interest.

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
