# Peer review of "The Effect of Boreal Summer Intraseasonal Oscillation on Evaporation Duct and Electromagnetic Propagation over the South China Sea"

_atmosphere, doi:10.3390/atmos11121298_

Round 1

Reviewer 1 Report

Title Suggestion: Boreal Summer Intraseasonal Oscillation impact on Evaporation Duct and Electromagnetic Propagation over the South China Sea

General question:

  • Did you check the influence of MJO and Equatorial Easterly Waves on evaporation duct over South China Sea (SCS)?
  • What human activities do impact the evaporation duct and electromagnetic waves?
  • Do you think the artificial island in SCS have a significant impact on evaporation duct and electromagnetic waves? Have you analyzed the data before and after the construction of the artificial island?

Comments and suggestions:

  • Line 71:- …. (NOAA) datasets.
  • Line 77:- the abbreviation TRMM must be written in all as Tropical Rainfall Measurement Mission (TRMM). And also, the dataset must be cited properly.
  • Line 78-79:- The Chinese Navy data should be cited properly with appropriate website (if applicable)
  • Line 91:- … mainland China or Chinese?
  • Line 94:- …. that lifecycle of the BSISO is around 40 days…. Is it? From the data I see between 50 and 60. Please clarify?
  • Line 193:- It is better to define Kuroshio current
  • Line 192-195:- Are these your findings or you are paraphrasing from previous findings? If you found them from other sources there need to be appropriate citation.
  • Figure 4:- Since Y-axis represent similar measurement for (a) & (b) using comparable scale (i.e. 0-21) helps the reader to see the difference
  • Line 288:- what are Figure a, b, and c represents?
  • Line 447:- What other factors you are referring to?
  • Line 500:- Briefly define electromagnetic waves
  • Line 512:- Section 6.2 or Observational Data should be in section 2
  • Line 523:- use the year 2018 instead of 2018
  • Figure 12 – Consider using color palette that can be easy to visualize by color blind people

Author Response

Dear Reviewer:

Thank you very much for your valuable suggestions. I  have read your  review comment carefully and  thought meaningfully about your comments. I will elaborate on  these points in the following paragraphs, inappropriate please criticism.

  1. General question

Point 1: Did you check the influence of MJO and Equatorial Easterly Waves on evaporation duct over South China Sea (SCS)?

Response 1: I'd like to talk about my points. Firstly, in Figure 3 and Figure 4, I’ve shown that the variance of BSISO-related sea surface evaporation is much higher during summer over the SCS. BSISO is  the dominant northward propagating component of MJO in summer, which is mostly active  in the Asian summer monsoon region, including the SCS. So I think that the effect of BSISO on evaporation duct can represent the major characteristic of MJO. Secondly, as for Equatorial Easterly Waves, I've looked through related articles since I have no experience with them. Equatorial Easterly Waves are the important tropical weather systems in summer, which can lead to wide range of convection weather. Evaporation duct is very sensitive to hydrometeorological factors at the air-sea boundary, so I believe that Equatorial Easterly Waves can definitely affect the duct. That's an excellent proposal, and I will cover this point in the next doctoral dissertation. Limited by the length of the article, I plan to  elaborate this point in Section 7.

Point 2: What human activities do impact the evaporation duct and electromagnetic waves?

Response 2: As far as I know, evaporation duct is a surface duct caused by evaporation from the sea surface, and is mainly formed in the open sea. Generally human activities are scarce in these areas, and the variations of evaporation duct caused by human activities can be ignored.

Point 3: Do you think the artificial island in SCS have a significant impact on evaporation duct and electromagnetic waves? Have you analyzed the data before and after the construction of the artificial island?

Response 3: As I’m still the officer of Chinese Navy, I can only talk briefly about this point due to some confidentiality and political reasons. Compared to the broad ocean, the artificial island appears to be negligible. Unlike the holistic man-made island, we just expanded the outer reef and enclose part of the bay to form a harbor. At ordinary times the population in artificial island is very sparse, and there are few  high-rise buildings. Unlike the the urban heat island effect, the artificial island in SCS has little impact on meteorological factors. As the observations of evaporation duct in SCS are extremely rare, I can only analyze the conventional meteorological data (wind, pressure,temperature,etc). The results showed no significant change after the construction of the artificial island.

  1. Comments and suggestions

Point 1: Line 71:- …. (NOAA) datasets.

Response 1: Thank you for pointing out my mistake. I will correct it immediately.

Point 2: Line 77:- the abbreviation TRMM must be written in all as Tropical Rainfall Measurement Mission (TRMM). And also, the dataset must be cited properly.

Response 2: Thank you for pointing out my mistake. I will correct it immediately.

Point 3: Line 78-79:- The Chinese Navy data should be cited properly with appropriate website (if applicable)

Response 3: I’m sorry that this data can not be cited properly. Because the data is undisclosed and I got it from a special channel.

Point 4: Line 91:- … mainland China or Chinese?

Response 4: I think “mainland China” is suitable in this sentence. I will correct it immediately.

Point 5: Line 94:- …. that lifecycle of the BSISO is around 40 days…. Is it? From the data I see between 50 and 60. Please clarify?

Response 5: Thank you for pointing out my mistake. I will correct it immediately.

Point 6: Line 193:- It is better to define Kuroshio current

Response 6: Thank you for pointing out my negligence. I will define it with reference.

Point 7: Line 192-195:- Are these your findings or you are paraphrasing from previous findings? If you found them from other sources there need to be appropriate citation.

Response 7: Thank you for pointing out my negligence. In the initial release, I drew the climatology of sea surface evaporation figure. In the process of revising I decided to delete this figure, because I thought it was not very important for the paper. Due to carelessness I forgot to revise relevant content in the main text. I will correct it immediately.

Point 8: Figure 4:- Since Y-axis represent similar measurement for (a) & (b) using comparable scale (i.e. 0-21) helps the reader to see the difference

Response 8: Thanks for your suggestions. I've thought about it before, but the Figure 4b was  inaesthetic when used comparable scale (i.e. 0-21). As far as I think, the power spectra figures are mainly drawn to show the time range of life-cycle, so the spectral scale may be less important. If you don't think it's appropriate, I can correct it immediately.

Point 9: Line 288:- what are Figure a, b, and c represents?

Response 9: I’m sorry that I didn’t make it clear in the legend. Figure 5a/b/c correspond to the first/second/third EOF modes in Section 2. I will comment it in the legend to make it even clearer.

Point 10: Line 447:- What other factors you are referring to?

Response 10: As far as I know, there are still some factors can affect the air humidity, such as temperature, wind, weather systems of different scales such as typhoon.

Point 11: Line 500:- Briefly define electromagnetic waves

Response 11: Thank you for pointing out my negligence. I will define it in the text.

Point 12: Line 512:- Section 6.2 or Observational Data should be in section 2

Response 12:  Thank you for pointing out my negligence. This part is really inappropriate to be a a separate section. So I plan to merges it into the next section.

Point 13: Line 523:- use the year 2018 instead of 2018

Response 13: Thank you for pointing out my mistake. I will correct it immediately.

Point 14: Figure 12 – Consider using color palette that can be easy to visualize by color blind people

Response 14:  Thank you for pointing out my negligence. I will change the color palette by deleting the green and grey color, so that it is easy to visualize by color blind people.

Reviewer 2 Report

This is the review comment of the manuscript entitled ‘The effect of boreal summer intraseasonal oscillation on evaporation duct and electromagnetic propagation over the South China Sea’ by Wentao Jia, Weimin Zhang, Jiahua Zhu, and Jilin Sun.

[Abstract]

This work investigates the relationship between the activity of the boreal summer intraseasonal oscillation (BSISO) in the South China Sea (SCS) and the evaporation duct that appears at the lower troposphere. The existent height and strength of this duct are enhanced (suppressed) in the negative (positive) phase of the BSISO activity. The less convection increases solar radiation and strengthens the heat flux from the sea surface during the negative phase of BSISO activity, which leads to an enhanced evaporation duct. Simulated electromagnetic fields are affected with the condition of the evaporation duct.

[Comments]

The manuscript is very well organized and the quality of the observation and analysis is sufficient. I think this study can be accepted for publishing this journal after some minor revisions. Some specific comments are as follows.

  1. Figure 3: The climatology of sea surface evaporation figure may be hidden.
  2. Figure 4: The spectral peeks of OLR and evaporation duct are different. The former is 40 days, while the latter is 50 days. Why is this difference occurred?
  3. Figure 5: Which figures correspond to the first/second/third EOF modes?
  4. Figure 12: Which column does correspond to active BSISO event?

Author Response

Dear Reviewer:

Thank you very much for your valuable suggestions. I  have read your  review comment carefully and  thought meaningfully about your comments. I will elaborate on  these points in the following paragraphs, inappropriate please criticism.

Point 1: Figure 3: The climatology of sea surface evaporation figure may be hidden.

Response 1: Thank you for pointing out my mistake. In the initial release, I drew the climatology of sea surface evaporation figure. In the process of revising I decided to delete this figure, because I thought it was not very important for the paper. Due to carelessness I forgot to revise relevant content in the main text. I will correct it immediately.

Point 2: Figure 4: The spectral peeks of OLR and evaporation duct are different. The former is 40 days, while the latter is 50 days. Why is this difference occurred?

Response 2: I'd like to talk about my points. Firstly, the physical mechanism of the evaporation duct is very complex and influenced by many factors. When influenced by other stronge weather systems, such as typhoon, the role of the BSISO may be less significant. These are mentioned at the end of the manuscript. Besides you can also find that there is a relative weak spectral peek of evaporation duct in 35 days. So I think it is reasonable that the spectral peeks of OLR and evaporation duct are not in full accord. Secondly, the power spectra figures may have more cues for the range of change cycle, that is the life cycle of 30-60 days.

Point 3: Figure 5: Which figures correspond to the first/second/third EOF modes?

Response 3: I’m sorry that I didn’t make it clear in the legend. Figure 5a/b/c correspond to the first/second/third EOF modes in Section 2. I will comment it in the legend to make it even clearer.

Point 4: Figure 12: Which column does correspond to active BSISO event?

Response 4: I’m sorry that I didn’t make it clear in the legend. Figure 12a and 12c (left column) correspond to active BSISO event, and the right correspond to negative BSISO. I will comment it in the legend to make it even clearer.
